# Nc886, a Novel Suppressor of the Type I Interferon Response Upon Pathogen Intrusion

**DOI:** 10.3390/ijms22042003

**Published:** 2021-02-18

**Authors:** Yeon-Su Lee, Xiaoyong Bao, Hwi-Ho Lee, Jiyoung Joan Jang, Enkhjin Saruuldalai, Gaeul Park, Wonkyun Ronny Im, Jong-Lyul Park, Seon-Young Kim, Sooyong Shin, Sung Ho Jeon, Sangmin Kang, Hyun-Sung Lee, Ju-Seog Lee, Ke Zhang, Eun Jung Park, In-Hoo Kim, Yong Sun Lee

**Affiliations:** 1Division of Clinical Cancer Research, Research Institute, National Cancer Center, Goyang 10408, Korea; yslee2@ncc.re.kr (Y.-S.L.); gaeulpark91@ncc.re.kr (G.P.); 2Department of Pediatrics, University of Texas Medical Branch, Galveston, TX 77555, USA; xibao@UTMB.EDU (X.B.); kezhang@utmb.edu (K.Z.); 3Department of Cancer Biomedical Science, Graduate School of Cancer Science and Policy, National Cancer Center, Goyang 10408, Korea; hhlee@ncc.re.kr (H.-H.L.); 74818@ncc.re.kr (J.J.J.); 91634@ncc.re.kr (E.S.); wk.ronny.im@gmail.com (W.R.I.); sangminkang@jbnu.ac.kr (S.K.); ejpark@ncc.re.kr (E.J.P.); ikim@ncc.re.kr (I.-H.K.); 4Department of Life Science and Multidisciplinary Genome Institute, Hallym University, Chuncheon 24252, Korea; ccyzzo@hallym.ac.kr (S.S.); sjeon@hallym.ac.kr (S.H.J.); 5Personalized Genomic Medicine Research Center, KRIBB, Daejeon 34141, Korea; nlcguard@kribb.re.kr (J.-L.P.); kimsy@kribb.re.kr (S.-Y.K.); 6Department of Functional Genomics, University of Science and Technology, Daejeon 34113, Korea; 7Division of Thoracic Surgery, Michael E. DeBakey Department of Surgery, Baylor College of Medicine, Houston, TX 77030, USA; Hyun-Sung.Lee@bcm.edu; 8Department of Systems Biology, University of Texas MD Anderson Cancer Center, Houston, TX 77030, USA; jlee@mdanderson.org

**Keywords:** nc886, pathogen, interferon, Protein Kinase R, Interferon Regulatory Factor 3

## Abstract

Interferons (IFNs) are a crucial component in the innate immune response. Especially the IFN-β signaling operates in most cell types and plays a key role in the first line of defense upon pathogen intrusion. The induction of IFN-β should be tightly controlled, because its hyperactivation can lead to tissue damage or autoimmune diseases. Activation of the IFN-β promoter needs Interferon Regulatory Factor 3 (IRF3), together with Nuclear Factor kappa-light-chain-enhancer of activated B cells (NF-κB) and Activator Protein 1 (AP-1). Here we report that a human noncoding RNA, nc886, is a novel suppressor for the IFN-β signaling and inflammation. Upon treatment with several pathogen-associated molecular patterns and viruses, nc886 suppresses the activation of IRF3 and also inhibits NF-κB and AP-1 via inhibiting Protein Kinase R (PKR). These events lead to decreased expression of IFN-β and resultantly IFN-stimulated genes. nc886′s role might be to restrict the IFN-β signaling from hyperactivation. Since nc886 expression is regulated by epigenetic and environmental factors, nc886 might explain why innate immune responses to pathogens are variable depending on biological settings.

## 1. Introduction

Interferons (IFNs) are cytokines that play a critical role in the host defense against pathogen intrusion (reviewed in [1]). By acting in an autocrine or paracrine manner, IFNs establish cellular anti-microbial states, limit pathogen spreading, and transmit signals to immune cells for mounting innate and adaptive immunity. The rise and demise of the IFN response should be tightly controlled, because its dysregulation can lead to deleterious outcomes such as infectious diseases, tissue damage, and autoimmune diseases. It awaits further investigation to elucidate how IFNs are differentially regulated depending on pathogens, characteristics of human individuals, pathological conditions, etc.

There are several IFNs that are classified into types I, II, and III based on sequence homology [2]. Among them, IFN-β, which is a member of type I IFNs and produced by most cell types in the body, plays a crucial in the first line defense against pathogens. The IFN-β signaling pathway is well characterized and provides a prototype how pathogen sensing results in transcriptional reprogramming (reviewed in [3]).

IFN-β is encoded by a single gene, *IFNB1*, and its transcription is activated when various cellular receptors sense microbial products (see Figure 1A, Figure 2A, and Figure 4B). For example, double-stranded RNA (dsRNA) or 5′-triphosphorylated RNA is recognized as non-self RNA by cytoplasmic sensors such as Retinoic Acid Inducible Gene I (RIG-I; DDX58 as an official name), Melanoma Differentiation-Associated Gene 5 (MDA5; IFIH1) or Protein Kinase R (PKR; EIF2AK2). In addition, there exists a group of membrane-bound sensors called Toll-Like Receptors (TLRs). TLR3 senses dsRNA, which is a viral genome itself or products of viral transcription or replication. TLR4 senses lipopolysaccharide (LPS), a membrane component of gram-negative bacteria [4]. Thus, dsRNA and LPS represent viral and bacterial infection respectively. Upon sensing, the signal is transduced through a number of molecular interactions, leading to phosphorylation of Interferon Regulatory Factor 3 (IRF3). Phospho-IRF3 translocates into the nucleus, binds to the IFN-β promoter region, and activates IFN-β transcription with the aid of Nuclear Factor kappa-light-chain-enhancer of activated B cells (NF-κB) and activator protein 1 (AP-1) (see Figure 2A, reviewed in [5]). After an infected cell produces and secrets IFN-β, it binds to the type I IFN receptor (IFNAR) to provoke the Janus kinase (JAK) and signal transducer and activator of transcription (STAT) pathway (reviewed in [6]). The consequence is transcriptional activation of a set of genes, called “interferon-stimulated genes (ISGs)” whose promoters have IFN-stimulated response elements (ISRE).

PKR has long been known as a key component of cellular innate anti-viral response (reviewed in [8]). Upon binding to non-self RNAs, typically dsRNA, PKR is auto-phosphorylated. Phospho-PKR is an active form and provokes diverse cellular signaling pathways including NF-κB and the mitogen-activated protein kinase (MAPK). The endpoint of the MAPK pathway is activation of the AP-1 transcription factor. PKR’s role in IFN-β expression is presumed by the fact that NF-κB and AP-1 bind to the IFN-β promoter (see Figure 2A and [3]). Indeed, there is experimental evidence that PKR is required for full activation IFN-β [9,10].

Our laboratory identified a human cellular non-coding RNA (ncRNA), nc886 (a.k.a, vtRNA2-1 or pre-miR-886) and found that one of its roles is to prevent PKR from being activated [11]. nc886 is a potent PKR regulator, owing to its binding affinity to PKR and cellular copy number. nc886′s K_D_ for PKR is ~12 nM and comparable to ~4 nM of dsRNA that is PKR’s best canonical activating ligand [12]. In addition, nc886 is abundantly expressed; its cellular level is 10^5^ RNA molecules per cell [11]. All aforementioned facts implicate nc886 in the IFN-β signaling. Here, we present our investigation on nc886′s novel role in the IFN-β signaling as a suppressor and further in the host defense against pathogens.

## 2. Results

### 2.1. nc886 Antagonizes the IFN-β Signaling

We examined a role of nc886 in the IFN-β signaling, given its importance in infection, inflammation, and associated disorders [13]. Pathogen intrusion triggers the expression of IFN-β that acts in an autocrine or paracrine manner, leading to activation of a set of promoters containing an ISRE(s) (shortly termed “ISRE promoter”) and ultimately to expression of ISGs (Figure 1A and Introduction).

We had constructed a 293T-derivative cell line that stably expressed nc886 (designated “293T:nc886”) as well as a control cell line which was isogenic except for nc886 (“293T:vector”) (Figure 1B and [11]). We first measured the expression level of IFN-β, a key gene in this signaling pathway. Transfection of Polyinosinic:polycytidylic acid (shortly “Poly(I:C)”), which is a synthetic dsRNA that mimics viral infection [14], into 293T cells increased the IFN-β mRNA level and this increase was attenuated when nc886 was present (293T:nc886 as compared to 293T:vector) (Figure 1C). To test whether this was attributed to the IFN-β promoter activity, we performed luciferase reporter assays. The luciferase assay yielded a similar result to the mRNA measurement, proving that nc886 suppressed IFN-β expression by acting on the IFN-β promoter (Figure 1D). Importantly, the nc886′s suppressive activity on the IFN-β promoter was also seen when it was activated by viral infection. Infection of Sendai virus (SeV) and respiratory syncytial virus (RSV) induced the IFN-β promoter, and this induction was weaker in nc886-expressing cells (Figure 1D). ISRE promoter activation, which is subsequent to the IFN-β induction, was also suppressed by nc886 (Figure 1E).

Next, we assessed nc886′s effect on the expression of ISGs. We retrieved and curated a list of 378 ISGs from a report in which they collected type I IFN-inducible genes from a number of papers ([15] and references therein). We examined those 378 ISGs in gene expression profiling data that we had obtained upon knockdown (KD) of nc886 in three esophageal cell lines; Het-1A, TE-1, and TE-8 [7]. From the 378 ISGs (actually corresponding to 594 probes, because one gene could have multiple probes), we sorted ISGs whose expression levels were significantly altered upon nc886 KD in all the three cell lines. A plot of their expression values showed that those ISGs tended to be higher in nc886 KD than control (designated “KD” and “ctrl”, respectively, in Figure 1F), providing evidence for a suppressive effect of nc886 on ISGs.

We also measured the production of CCL5 (a.k.a. RANTES), one of the ISGs whose induction needs IFN-β [16]. An enzyme-linked immunosorbent assay (ELISA) showed that viral infection induced the production of RANTES, to a less extent in 293T:nc886 than 293T:vector cells (Figure 1G). All these data provided clear evidence that nc886 plays a suppressive role in the IFN-β signaling cascade.

### 2.2. nc886 Suppresses the IFN-β Promoter Activity via Inhibiting PKR

The IFN-β promoter contains three regulatory elements; AP-1, IRF, and NF-κB (Figure 2A and [3]). PKR is an upstream activator of AP-1 and NF-κB [8] and so is implicated in the IFN-β signaling. Previous reports showed that the induction of IFN-β expression by synthetic non-self RNA or a virus is attenuated upon PKR KD by short hairpin RNA or short interfering RNA (siRNA) [9,10]. We corroborated these KD data in a PKR knockout (KO) cell line that we had generated from an immortalized thyroid cell line, Nthy-ori 3-1 (Figure 2B). The induction of IFN-β expression by Poly(I:C) was less in PKR KO cells than in the parental wild type (wt) cells (Figure 2C). Our data ensured the requirement of PKR in the IFN-β signaling.

nc886 is a PKR repressor [11,12], which was reassured here by lower PKR activity in nc886-expressing cells than non-expressing ones (Appendix A). Based on this, we reasoned that PKR could explain nc886′s suppressive effect on the IFN-β signaling. To prove this, we performed PKR KD in nc886-sufficient or deficient cells and measured the IFN-β and ISRE promoter activity (Figure 2D–G). In the pair of 293T cells, nc886 suppressed both promoters in the presence of PKR (see the left “siControl” part in Figure 2E), which is consistent with our Figure 1 data. Importantly, this nc886-dependent difference disappeared mostly when PKR was depleted (compare 293T:vector and 293T:nc886 in the right “siPKR” part in Figure 2E). These data proved that nc886′s effect was dependent on PKR.

We also conducted nc886 KD experiments in the HCT116 cell line that naturally expresses nc886 [11]. nc886 KD was sufficient to induce the IFN-β and ISRE promoter activity (Figure 2F–G). This induction in the presence of PKR was significantly higher than that upon PKR KD (compare plain bars between “siControl” and “siPKR” in Figure 2G). All these data convincingly demonstrated that nc886′s suppressive effect on the IFN-β signaling was through inhibiting PKR.

### 2.3. nc886 Restricts the IFN-β Promoter Activation by Inhibiting NF-κB and/or AP-1 Pathways

We further investigated each of the three elements in the IFN-β promoter (NF-κB, AP-1 and IRF3; see Figure 2A). We first looked into NF-κB and AP-1, which are PKR downstream pathways. To evaluate nc886′s effect on these two elements, we employed IRF3(5D), a constitutively active phospho-mimetic mutant, which is known to activate the IFN-β promoter when ectopically expressed. Since this activation bypasses all signaling events upstream of IRF3, any difference caused by nc886 in this context should be attributed to NF-κB and/or AP-1. Transfection of an IRF(3D)-expressing plasmid into the pair of 293T cells activated the IFN-β promoter, and this activation was mitigated in the presence of nc886 (Figure 3A). The mitigation was modest as compared to that of Poly(I:C) that was shown as a positive control (left panel of Figure 3A) but was at a statistically significant level.

Having confirmed that nc886 suppressed NF-κB and/or AP-1, we further scrutinized individual elements. To assess the NF-κB activity, we measured expression levels of two representative NF-κB target genes, *NFKBIA* and *TNFAIP3*. Their expression was induced by Poly(I:C), and this induction was less in nc886-expressing cells (Figure 3B). nc886′s effect on NF-κB activity was also evaluated by a luciferase assay measuring the activity of a promoter containing NF-κB responsive elements. This promoter was induced by viral infection, and this induction was attenuated in nc886-expressing cells (Figure 3C). These data proved nc886′s suppressive effect on the NF-κB pathway.

Next, we interrogated whether nc886 suppressed AP-1 activation by measuring FOS, a component of the AP-1 transcription factor for IFN-β activation [17,18]. FOS phosphorylation represents its active form. nc886 KD lead to FOS phosphorylation in PKR-sufficient cells, but not in PKR-KO cells (Figure 3D). This result validated that nc886 suppressed AP-1 activation in a PKR-dependent manner.

Our data collectively demonstrated that nc886 repressed NF-κB and AP-1 pathways and that this repression was one reason why the IFN-β signaling was weaker in nc886-expressing cells.

### 2.4. nc886 Inhibits IRF3 via the RIG-I/MAVS Pathway

Close inspection of our data indicated a possibility that nc886 inhibited not only NF-κB and AP-1 but also IRF3. First, we consistently observed that nc886 was more suppressive to the IFN-β promoter when triggered by Poly(I:C) than by IRF3(5D) (for direct comparison, see Figure 3A). An alteration of the IFN-β promoter activity should be attributed to all the three elements in the case of Poly(I:C), but most probably to only two elements (NF-κB and/or AP-1) in the case of IRF3(5D). Second, nc886 KD by itself was sufficient to activate the IFN-β promoter (Figure 2G). In contrast, activation of the NF-κB pathway by expression of p65 was insufficient (Appendix A). Thus, we sought to determine a role of nc886 in the IRF3 pathway.

To measure the activity of IRF3, we performed luciferase assays by using a reporter plasmid harboring four copies of the IRF3-binding motif. This promoter was activated by Poly(I:C), and this activation was diminished by nc886 (Figure 4A). As a control experiment, we activated this promoter by expressing IRF3(5D). As expected, transfection of the IRF3(5D)-expressing plasmid led to a robust induction equally well between nc886-expressing cells and non-expressing ones. In other words, nc886 was incapable of exerting any effect on the phospho-mimetic form, suggesting that nc886 would have acted on a step(s) prior to IRF3 phosphorylation.

IRF3 is activated from multiple pathways. Upon viral infection, non-self RNAs are sensed by two representative pathogen recognition receptors (reviewed in [19]). These two are cytoplasmic RIG-I (or MDA5) and membrane bound TLR3, which transduce signal via Mitochondrial AntiViral-Signaling protein (MAVS) and TIR-domain-containing adapter-inducing interferon-β (TRIF), respectively, to merge at TANK-binding kinase 1 (TBK1) and IκB kinase ε (IKKε) (Figure 4B for simplified depiction). Ectopic expression of RIG-I, MAVS, or TLR3 elicited activation of the IFN-β promoter (Figure 4C). nc886 suppressed the IFN-β promoter when it was induced by RIG-I or MAVS but did not when induced by TLR3. Ectopic expression of MAVS led to IRF3 phosphorylation, which was attenuated when nc886 was co-expressed (Figure 4D). Our data indicated that nc886 suppressed IRF3 and that nc886 targeted a stage(s) prior to TBK1/IKKε in the RIG-I/MAVS pathway.

### 2.5. nc886 also Inhibits Inflammatory Genes in Macrophages

Macrophages play a paramount role in the inflammatory process. Upon sensing bacterial infection, they release cytotoxic molecules and cytokines, such as nitric oxide (NO) and interleukin 6 (IL6) (reviewed in [4,20]). We investigated a role of nc886 in IFN-β and inflammation of macrophages. RAW 264.7 is a mouse macrophage cell line and does not express nc886. We constructed a RAW 264.7 derivative cell lines which stably expressed nc886 or a control cell line which did not express nc886 (designated “RAW:nc886” and “RAW:vector” in Figure 5A). These cells were treated with LPS, a molecular pattern of gram-negative bacteria. LPS treatment activated the IFN-β promoter also in macrophages and this activation was weaker when nc886 was expressed, as shown by luciferase assays (Figure 5B). Next, we measured the expression of *Il6* and *Nos2*. *Nos2* encodes an enzyme “Nitric oxide synthase, inducible”, which is responsible for NO production by macrophages upon pathogen intrusion [20]. LPS treatment induced *Il6* and *Nos2* expression, and this induction was remarkably diminished in RAW:nc886 cells as compared to RAW:vector cells (Figure 5C). Consequently, nc886 inhibited IL6 secretion and NO production which were induced by LPS (Figure 5D). All these data indicated that nc886 suppressed the IFN-β signaling also in macrophages and attenuated their response to pathogens.

## 3. Discussion

Our study here has determined nc886′s role in the IFN-β signaling. We found nc886 to suppress IRF3 most likely through the RIG-I/MAVS pathway. Our data corroborated nc886′s repressive effect on NF-κB and AP-1 via PKR inhibition. Consequently, nc886 is a regulator for ISG expression and the host anti-viral response. This regulation is expected to operate in most cell types, since nc886 is transcribed by RNA polymerase III and thus is present ubiquitously in normal tissues [11,21]. In addition, we have shown that nc886 suppresses LPS-induced inflammatory response in a murine macrophage cell line.

Regarding the regulation of IRF3, nc886 inhibited the signaling pathway from RIG-I and/or MAVS. Based on our luciferase assay data that nc886 did not affect the TLR3 pathway (Figure 4B–C), we surmised that nc886 would target a stage(s) prior to TBK1/IKKε in the signaling network for IRF3 activation. Like PKR, RIG-I is a RNA binding protein and pathogen sensor. nc886 competes with intruder RNAs and thereby inhibits PKR. Therefore, an intuitively plausible possibility is that nc886 might bind to RIG-I and interfere with its binding to intruder RNAs. However, our data did not support this scenario. Previously, we conducted biochemical screening to identify proteins that are physically associated with nc886 [11,22] but failed to see RIG-I in the mass spectrometry data. In an alternative approach, we did immunoprecipitation (IP) experiments to capture RIG-I and sought to detect nc886 therein. nc886 was not detectable in the RIG-I IP complex but was reproducibly present in the PKR-IP complex as a positive control (data not shown). RIG-I is unlikely to be a direct molecular target of nc886. PKR is also unlikely to account for nc886′s effect on IRF3, because a previous study showed that PKR KD did not affect IRF-3 activation [9,10]. The molecular mechanism how nc886 suppresses IRF3 awaits further study.

Our result is intriguing, because nc886, albeit being a host ncRNA, appears to be antagonistic to host cells in a battle against pathogens. As we mentioned in Introduction, the activity of IFNs should be tightly controlled. IFN-β activation is essential for protection of an organism from invading pathogens but inevitably accompanies apoptotic cell death and recruitment of cytotoxic immune cells. nc886′s role might be to restrict IFN-β activity under a certain threshold level, to prevent collateral tissue damage. As important as IFN-β activation upon pathogen intrusion is its inactivation after host cells subdue pathogens. nc886 might play a role during healing of host tissues after the battle comes to an end. Concerning these possibilities, there is an interesting report in which the expression of nc886 (referred as CBL-3, an alias of nc886, in the paper) was stimulated by Epstein–Barr virus (EBV) [23]. We surmise that the induced nc886 could play two different roles. nc886 might be important in quelling IFN-β when EBV is cleared by host’s antiviral action. Alternatively, EBV might exploit nc886 for its replication. In any case, nc886 adds another layer of control in host-pathogen interaction.

Besides EBV infection, nc886 expression is dynamically regulated in diverse biological settings [21,24]. Our findings here indicate that the IFN response becomes different according to the nc886 expression level. The susceptibility to pathogens and clinical outcomes vary among individuals and also depending on a status of each individual. However, the underlying mechanism has yet to be elucidated. In this regard, it is interesting to cite several reports about nc886. The expression level of nc886 could be different from one person to another [25]. nc886 expression is dysregulated in pathological conditions, especially in cancer [26]. There might be a link between nc886 and IFN, since IFN has also an anti-tumor role besides its anti-pathogenic role and was once considered as a promising cancer therapeutic drug [27]. In conclusion, we envision that nc886 might elucidate the unsolved questions about the IFN response and susceptibility to pathogens and could provide future clinical utility.

## 4. Materials and Methods

### 4.1. Cell Lines, Viruses, Antibodies, and Other Reagents

HEp-2 cells were from American Type Culture Collection (Manassas, VA, USA). 293T, HCT116, Huh7, and RAW264.7 cell lines were our laboratory stock. 293T:nc886 and 293T:vector cell lines are described in [11]; Nthy-ori 3-1 wild type (wt) and PKR knockout (KO) cell lines are in [28]. From the RAW264.7 cell line, we generated an nc886 expressing cell line by transfecting an nc886-expressing plasmid, named “pCAGGS-GFP/886”. This plasmid was constructed from pCAGGS-GFP, which had the green fluorescent protein (GFP)-encoding gene inserted into pCAGGS-Neo vector. We PCR-amplified 649 nucleotide long DNA harboring the nc886 gene and flanking sequences and inserted it into pCAGGS-GFP, to produce pCAGGS-GFP/886. These two plasmids were used to construct RAW:vector and RAW:nc886 respectively. Plasmid transfection was performed with Lipofectamine™ 2000 reagent (Invitrogen, Carlsbad, CA, USA) and selection of clones in the presence of G418 was done according to general laboratory procedures.

Sendai virus was purchased from Charles River Laboratories, Inc (Wilmington, MA, USA). Respiratory syncytial virus (RSV) is our laboratory stock and a long strain was propagated in HEp-2 cells at 37 °C and purified by centrifugation in sucrose gradient as described [29]. Viral titer was determined by immunostaining of infected HEp-2 cells using polyclonal biotin-conjugated goat anti-RSV antibody (cat #: 7950-0104, Bio-rad, Hercules, CA, USA) and streptavidin peroxidase polymer (Sigma-Aldrich, St Louis, MO, USA) sequentially, as previously described [30]. LPS was from Sigma-Aldrich.

Antibodies were purchased from Cell Signaling Technology (shortly “CST”, Danvers, MA, USA), Abcam (Cambridge, MA, USA), or Sigma-Aldrich. Information about each of them is as follows: total PKR (ab32052, Abcam), phospho-PKR (ab32036, Abcam), total FOS (#4384, CST), phospho-FOS (#5348, CST), total IRF3 (#11904, CST), phospho-IRF3 (#4947, CST), and β-actin (A1978, Sigma-Aldrich).

### 4.2. Luciferase Reporter Assays

The information about plasmids encoding the firefly (*Pp*) luciferase gene whose expression is driven by each promoter is as follows: the IFN-β promoter, [12]; the ISRE promoter, [31]; the NF-κB-responsive promoter, [11]; the IRF3-responsive promoter, [32]. Cells were transfected with an indicated *Pp*-plasmid together with pRL-SV40 (Promega, Madison, WI) that expresses *Renilla* luciferase (*Rr*), using FuGene 6 (Roche, Indianapolis, IN, USA) or Lipofectamine™ 2000 reagent (Invitrogen). Promoter activity was measured by Dual-Luciferase^®^ Reporter Assay System (Promega). *Pp* values normalized to *Rr* values were obtained from triplicate or quadruplicate samples. An average, a standard deviation, and a *p*-value calculated by Student’s T-test were displayed in graphs.

### 4.3. Plasmids, Synthetic RNAs, and Their Transfection

Information about plasmids expressing each gene is as follows: IRF3(5D), RIG-I, MAVS [33]; TLR3 (a kind gift from Dr. Kui Li at the University of Tennessee). The antisense oligonucleotides against nc886 and non-target control were described in [11]. “anti-nc886” and “anti-control” in the text and figures indicate “anti886 75_56” and “anti_vt 21_2”, respectively, in the reference. siRNAs directed against PKR and control were Stealth RNAi™ siRNA (Invitrogen), as described in [34]. Poly(I:C)-HMW(1.5–8 kb) and Poly(I:C)-LMW (0.2–1 kb) were purchased from Invivogen (San Diego, CA, USA). In most experiments, Poly(I:C)-LMW was used unless otherwise mentioned in figure legends. Plasmids and Poly(I:C) were transfected with Lipofectamine™ 2000 reagent (Invitrogen) per the manufacturer’s instructions. Anti-oligos and siRNAs were with Lipofectamine^TM^ RNAiMAX reagent (Invitrogen).

### 4.4. RNA Isolation and Measurement

Total RNA was isolated by TRIzol™ Reagent (Invitrogen). Northern hybridization and qRT-PCR was done as previously described [11]. qRT-PCR primers are listed below in a format of gene name (forward primer 5′ to 3′, reverse primer 5′ to 3′). *IFNB1* (gcagttccagaaggaggacg, tccagccagtgctagatgaatc); *18S rRNA* (cggctttggtgactctagat, gcgactaccatcgaaagttg); *TNFAIP3* (gaagcaccatgtttgaaggatactg, ctctgcgctggctcgatc); *NFKBIA* (aggagagctatgacacagagtcagag, tgcgctcataacgtcagacg); *Il6* (gccagagtccttcagagagataca, attggatggtcttggtccttagcc); *Nos2* (gctaccacattgaagaagctggtg, ccataggaaaagactgcaccgaag); *Gapdh* (gacatcaagaaggtggtgaagcag, ccctgttgctgtagccgtattcat).

### 4.5. ELISA and NO Measurement

Immunoreactive RANTES was measured as described previously [35], by using DuoSet ELISA kit (R&D Systems, Minneapolis, MN, USA). Immunoreactive IL6 was quantified by Mouse IL-6 ELISA Kit (Koma Biotech Inc., Seoul, Korea) per manufacturer’s protocol. The level of secreted NO was assessed by measuring nitrite with Griess Reagent System (Promega).

### 4.6. Statistical Analysis

We used paired (for ISG expression in Figure 1F) and unpaired Student’s t-test (for all other assays) to evaluate the significance using R software (version 3.4.0). For qRT-PCR and luciferase assays, *p*-values were calculated from triplicate samples unless specified otherwise in the figure legends. *p*-values were displayed by asterisks on graphs. One, two, and three asterisks designate a *p*-value less than 0.05, 0.01, and 0.001, respectively. “n.s.”: not significant.

## Figures and Tables

**Figure 1 ijms-22-02003-f001:**
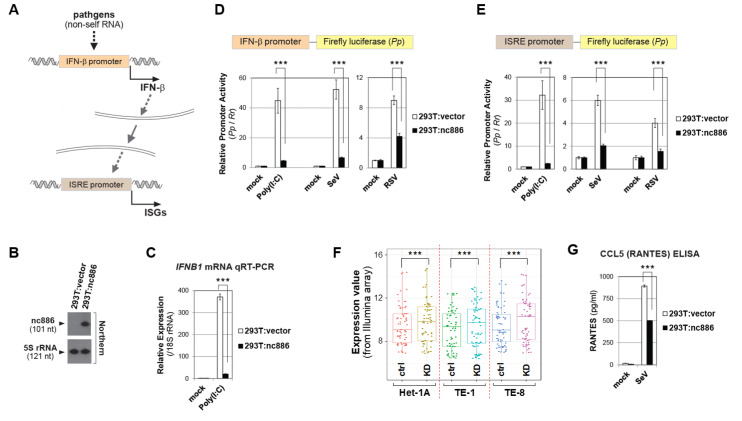
nc886 suppresses the IFN-β signaling. (**A**) Image showing the IFN-β signaling leading to ISG expression. See the text for details. (**B**) Northern hybridization of nc886 and 5S rRNA as a loading control in indicated cell lines. (**C**) qRT-PCR measurement of IFN-β mRNA (from the *IFNB1* gene) at 8 h after Poly (I:C) transfection. *y*-axis: 2^-ΔΔCt^ values normalized to 18S rRNA and relative to the value of mock transfected 293T:vector being set as 1. (**D**) Luciferase assays for the firefly (*Pp*) plasmid depicted above, upon experimental treatments indicated in the figure caption. A *Pp* value was normalized to the *Renilla* (*Rr*) luciferase value from co-transfected pRL-SV40. For each treatment, a *Pp*/*Rr* value relative to the corresponding “mock” value was plotted (*y*-axis). An average, a standard deviation, and a *p*-value were calculated from pentaplicate samples. Initial transfection of luciferase plasmids (time at 0 h); Poly(I:C)-LMW transfection or viral infection at 9 h; luciferase assays at 24 h. RSV and Sendai was infected at one multiplicity of infection (MOI). (**E**) ISRE luciferase assays. All the descriptions are the same as panel D, except for Poly(I:C)-LMW transfection and viral infection at 15 h. (**F**). Expression of selected ISGs upon nc886 KD (by an anti-oligo targeting nc886; designated as “KD”) and control KD (by a non-targeting oligo; “ctrl”). Details about KD experiments are in the reference [7]. Expression values (*y*-axis) are Illumina array values of 23 genes that were significantly altered in nine experimental sets (three cell lines, triplicates each). Paired t-test was used to calculate *p*-values between “ctrl” and “KD”. (**G)**. RANTES ELISA from the culture supernatant at 24 h after SeV infection. Three asterisks designate a *p*-value less than 0.001, respectively.

**Figure 2 ijms-22-02003-f002:**
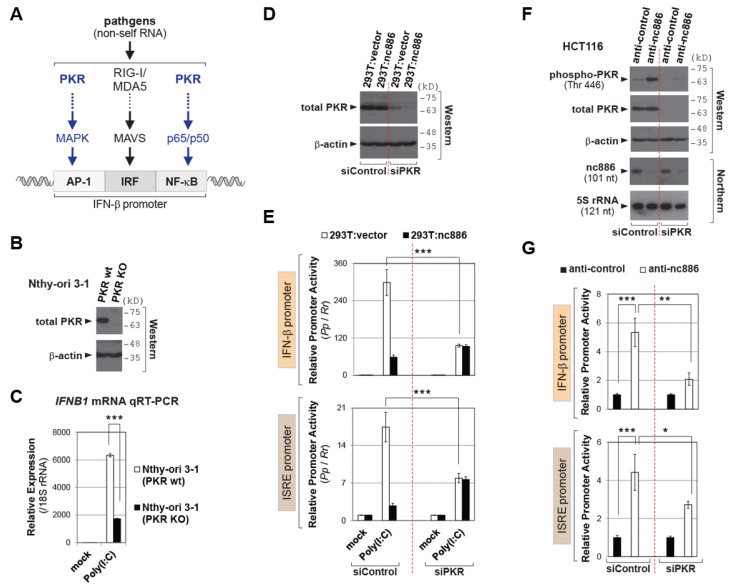
nc886 suppresses the IFN promoter via inhibiting PKR. (**A**) Image showing the IFN-β promoter as well as regulatory factors and pathways. (**B**) Western blot of PKR and β-actin as a loading control. (**C**) qRT-PCR as described in Figure 1C. 2^−ΔΔCt^ values were relative to the mock transfected PKR wt sample. (**D**) Western blot of PKR and β-actin as in panel B. (**E**) Luciferase assays for the reporter plasmid indicated on the left. Initial siRNA transfection (0 h); transfection of luciferase plasmids at 12 h, cell harvest and assays at 36 h. All other information is the same as Figure 1D. (**F**) Western and Northern blot of indicated proteins and RNAs, at 24 h after co-transfection of indicated siRNAs and anti-oligos. (**G**) Luciferase assays, as described in panel E and F. Luciferase plasmids were co-transfected with siRNAs and anti-oligos. Three asterisks designate a *p*-value less than 0.001, respectively.

**Figure 3 ijms-22-02003-f003:**
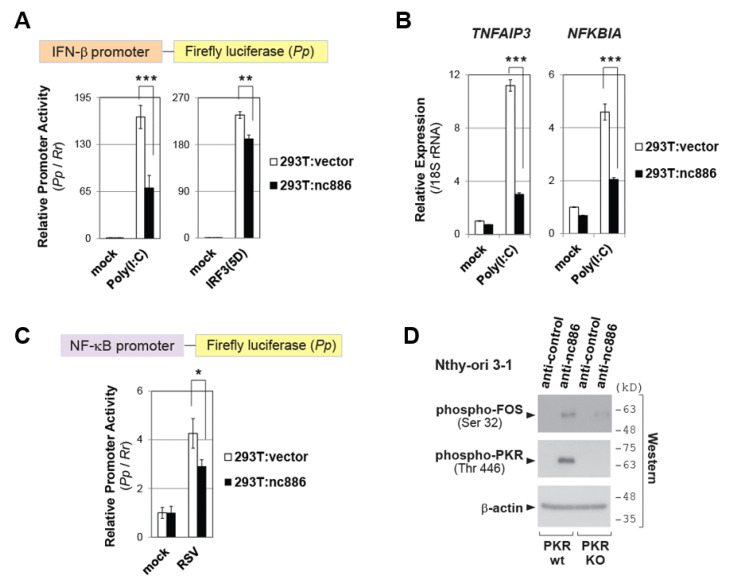
nc886 suppresses NF-κB and AP-1, two PKR downstream pathways. (**A**) Luciferase assays. Poly(I:C)-HMW was used. Poly(I:C) or an IRF3(5D)-expressing plasmid was co-transfected with luciferase plasmids, followed by cell harvest at 24 h. All the other descriptions are the same as Figure 1D. (**B**) qRT-PCR measurement of indicated NF-κB target genes, at 3 h after transfection of Poly(I:C). (**C**) Luciferase assays as described in Figure 1D. (**D**) Western blot of indicated proteins, as described in Figure 2B. Cells were harvest at 24 h after transfection of anti-oligos. One, two, and three asterisks designate a *p*-value less than 0.05, 0.01, and 0.001, respectively.

**Figure 4 ijms-22-02003-f004:**
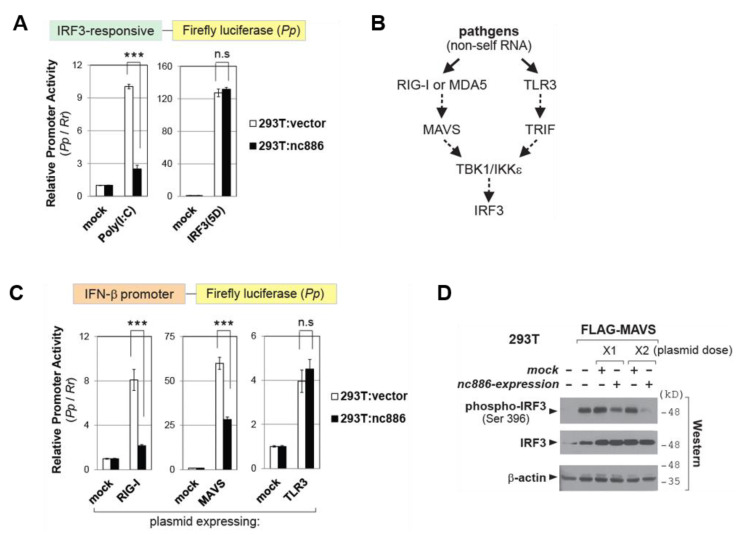
nc886 suppresses the IRF3 pathway. (**A**) Luciferase assays as described in Figure 3A, except that Poly(I:C)-LMW was used. (**B**) Simplified image for RIG-I and TLR3 pathways. Solid arrows designate pathogen sensing by indicated proteins; dotted arrows designate signal transduction between indicated proteins, which often involves multiple other factors. (**C**) Luciferase assays as described in panel A. (**D**) Western blot of indicated proteins at 24 h after transfection of MAVS-expressing plasmid in combination with control (“mock”) or nc886-expressing plasmid. Three asterisks designate a *p*-value less than 0.001. “n.s.”: not significant.

**Figure 5 ijms-22-02003-f005:**
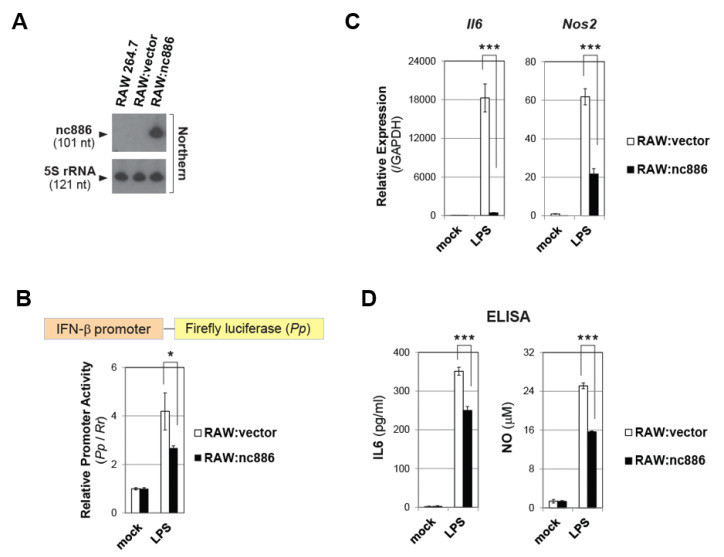
nc886 attenuates inflammatory responses in macrophages triggered by LPS. (**A**) Northern hybridization of nc886 and 5S rRNA as a loading control. (**B**) Luciferase assays. Transfection of luciferase plasmids at 0 h, treatment of LPS (10 ng/mL) at 6 h, and cell harvest and luciferase assays at 24 h. All other descriptions are the same as Figure 1D. (**C**) qRT-PCR of indicated genes at 6 h after treatment with LPS (100 ng/mL). (**D**) ELISA to measure secreted IL6 and NO, at 24 h after treatment with LPS (100 ng/mL). One asterisk designates a *p*-value less than 0.05; Three asterisks designate a *p*-value less than 0.001.

## Data Availability

Not applicable.

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
