# Peer review of "Nc886, a Novel Suppressor of the Type I Interferon Response Upon Pathogen Intrusion"

_ijms, 2021, doi:10.3390/ijms22042003_

Round 1

Reviewer 1 Report

please continue he  reseaches, it is of major practical and theoretical interest

Author Response

For easy reading, our responses are in brackets highlighted by four asterisks ****[….] to distinguish them from the original comments.

Reviewer 1 Comments

****[our responses]

please continue he  reseaches, it is of major practical and theoretical interest

****[Thanks for your time and also for the compliment]

Reviewer 2 Report

I read with very interest the draft entitled “nc886, a novel suppressor of the type I interferon response upon pathogen intrusion”. The paper well written and focused on the role of a new human cellular non-coding RNA,nc886 in the Interferons-β signaling. Authors report that nc886 is a regulator for the expression of interferon-stimulated genes and thus for host anti-viral response.

The manuscript is clearly written and the presented result are fully understandable, there are, however, some shortcomings that should be addressed properly.

  • Regarding the results, in description of Figure 1 Authors report the p value and the statistical test used. In other figure they report “n.s.: not significant”. Moreover, the symbol * is reported in graphs, indicating significantly differences. My suggestion is to add in Material and Methods a Statistical analysis paragraph for reporting the statistical test used for all data. Please, also indicate the version of statistical software used.
  • Regarding Figure 2 and 5, I suggest to improve the resolution of them, as it is difficult to understand.
  • The manufacturer’s information of the methods used have been usually mentioned alongside the material/instrument. This information includes the name of the supplier, state, and country. Example: serum creatinine was measured by the Architect C800 (Abbott Laboratories, Wiesbaden, Germany).
  • Finally, in order to
  • I suggest a conclusion section of your research paper including an overall summary and further research.
  • Finally, in order to more elucidate the results observed and the future perspectives about them,  I suggest completing with a conclusion paragraph

Therefore, I believe that the present paper should be accepted after minor revision.

Author Response

For easy reading, our responses are in brackets highlighted by four asterisks ****[….] to distinguish them from the original comments. 

Reviewer 2 Comments

****[our responses]

I read with very interest the draft entitled “nc886, a novel suppressor of the type I interferon response upon pathogen intrusion”. The paper well written and focused on the role of a new human cellular non-coding RNA,nc886 in the Interferons-β signaling. Authors report that nc886 is a regulator for the expression of interferon-stimulated genes and thus for host anti-viral response.

The manuscript is clearly written and the presented result are fully understandable, there are, however, some shortcomings that should be addressed properly.

****[Thanks for your time and also for the compliment]

Regarding the results, in description of Figure 1 Authors report the p value and the statistical test used. In other figure they report “n.s.: not significant”. Moreover, the symbol * is reported in graphs, indicating significantly differences. My suggestion is to add in Material and Methods a Statistical analysis paragraph for reporting the statistical test used for all data. Please, also indicate the version of statistical software used.

****[We have added a separate section “Statistical analysis” in Materials and Methods (line 385-391) and clearly described about all information (statistical methods, software/version, etc)]

Regarding Figure 2 and 5, I suggest to improve the resolution of them, as it is difficult to understand.

****[For all figures (not only Figures 2 and 5), we have replaced with high-resolution ones]

The manufacturer’s information of the methods used have been usually mentioned alongside the material/instrument. This information includes the name of the supplier, state, and country. Example: serum creatinine was measured by the Architect C800 (Abbott Laboratories, Wiesbaden, Germany).

****[I agree with the reviewer’s opinion. We indicated the manufacturer’s information according to the following rules: manufacturer name, city, state (in the case of manufacturers in United States); manufacturer name, city, country (for non-US countries). We included full information when a manufacturer appears in the text for the first time. From the second time and on, we omitted the location information. We double-checked and confirmed that all manufacturers’ information has been consistently described. We will comply with IJMS policy, if the journal has a different rule.]

I suggest a conclusion section of your research paper including an overall summary and further research.

Finally, in order to more elucidate the results observed and the future perspectives about them,  I suggest completing with a conclusion paragraph

****[We have included (line 308-320): “Besides EBV infection, nc886 expression is dynamically regulated in diverse biologi-cal settings {Park, 2017 #726}{Lee, 2015 #1092}. Our findings here indicate that the IFN re-sponse becomes different according to the nc886 expression level. The susceptibility to pathogens and clinical outcomes vary among individuals and also depending on a status of each individual. However, the underlying mechanism has yet to be elucidated. In this regard, it is interesting to cite several reports about nc886. The expression level of nc886 could be different from one person to another {Silver, 2015 #448}. nc886 expression is dysregulated in pathological conditions, especially in cancer {Lee, 2020 #1131}. There might be a link between nc886 and IFN, since IFN has also an anti-tumor role besides its anti-pathogenic role and was once considered as a promising cancer therapeutic drug {Arico, 2019 #1155}. In conclusion, we envision that nc886 might elucidate the unsolved questions about the IFN response and susceptibility to pathogens and could provide fu-ture clinical utility.”]

Therefore, I believe that the present paper should be accepted after minor revision.

****[I believe (and hope) that our revision is satisfactory and so that our manuscript be accepted]